# SCA-Safe Implementation of Modified SaMAL2R Algorithm in FPGA

**DOI:** 10.3390/mi13111872

**Published:** 2022-10-30

**Authors:** José de Jesús Morales Romero, Mario Alfredo Reyes Barranca, David Tinoco Varela, Luis Martin Flores Nava, Emilio Rafael Espinosa Garcia

**Affiliations:** 1Department of Electrical Engineering, CINVESTAV, Mexico City 07360, Mexico; 2Engineering Department, Superior Studies Faculty-Cuautitlán, National Autonomous University of Mexico, UNAM, Cuautitlán Izcalli 54714, Mexico

**Keywords:** RSA, Montgomery multiplication, Montgomery exponentiation, SCA, SPA, *N* − 1 *Attack*, SaMA, BRIP, FPGA

## Abstract

Cryptographic algorithms (RSA, DSA, and ECC) use modular exponentiation as part of the principal operation. However, Non-profiled Side Channel Attacks such as Simple Power Analysis and Differential Power Analysis compromise cryptographic algorithms that use such operation. In this work, we present a modification of a modular exponentiation algorithm implemented in programmable devices, such as the Field Programmable Gate Array, for which we use Virtex-6 and Artix-7 evaluation boards. It is shown that this proposal is not vulnerable to the attacks mentioned previously. Further, a comparison was made with other related works, which use the same family of FPGAs. These comparisons show that this proposal not only defeats physical attack but also reduces the number of resources. For instance, the present work reduces the Look-Up Tables by 3550 and the number of Flip-Flops was decreased by 62,583 compared with other works. Besides, the number of memory blocks used is zero in the present work, in contrast with others that use a large number of blocks. Finally, the clock cycles (latency) are compared in different programmable devices to perform operations.

## 1. Introduction

Nowadays, most of the confidential information and data of people, governments, and companies, are exchanged via electronic media. Electronic media have the advantage of increasing the speed of information exchange. Moreover, they supply an elevated level of security, confidentiality, and data integrity.

Many researchers have developed various methods for encryption and decryption of secret information, including methods with artificial intelligence, as in Ref. [1]. However, attackers have engaged in these techniques.

Encryption and decryption of the information use a secret key. Cryptographic devices such as Smart Cards, Field Programmable Gate Arrays (FPGA), Application-Specific Integrated Circuits (ASICs), microcontrollers, and others operate storing this secret key.

A device that is very interesting to use for various applications is the FPGA. These devices can be used in various fields, such as artificial intelligence [2,3], power electronics [4], control [5], optimization [6], among others. Due to the versatility of these devices, we use them, in the present work, within the area of cryptography.

Kocher, in the 1990s, realized that anyone could get the secret key through physical measurements in cryptographic devices [7,8].

These attacks do not require highly specialized equipment since the attack can be carried out using accessible equipment, such as an oscilloscope and resistors.

Hence, Kocher proved that physical attacks are possible and not only at the software (or algorithm) level. This type of attack is aimed at devices that anyone uses, such as those used in Internet of Things (IoT) [9,10]. These kinds of devices hold confidential information of the owners, such as email, bank movements, personal identification, and any kind of sensitive and private information.

Therefore, works have been focused on the implementation of secure algorithms. In order to reject or impede access to secret information, approaches to reach this task can be either based on software or hardware [9,10]. This last approach has the disadvantage of being very demanding in resource consumption.

This paper deals with modular exponentiation and its implementation to protect information from some of the Non-Profiled Side Channel Attacks [11]. We propose an algorithm that shows robustness, security, and reliability, compared with other reported implementations, analyzing, in particular, its performance upon the N−1Attack, which is a type of Non-Profile Side Channel Attack.

We will show that the present work reduces a minimum of 3559 Look-Up Tables (LUT) and 62,583 Flip-Flops (FF) compared to other architectures. In addition, it will be shown that the present work does not require memory blocks compared to the rest of the works cited in this one.

The rest of this work is as follows: Section 2 reviews the Side-Channel Attacks (SCA) in the context of RSA implementations; next, in Section 3, an absorbing attack called the N−1Attack is reviewed; later, in Section 4, the proposed implementation of modular exponentiation and modular multiplication is reviewed; afterward, in Section 5, the implementation of the proposed modular exponentiation and the modular Multiplication implemented in an FPGA is shown; finally, Section 6 provides the results of the proposed approach in the FPGA.

## 2. Side-Channel Attacks

Since the first physical assault described by Kocher, researchers have developed many attacks against programmable devices. These attacks are known as Side-Channel Attacks (SCA). In the literature, there are two main categories for SCAs: the first category is related to profile attacks, and the second is related to non-profile attacks [11,12].

### 2.1. Profiled SCAs

Profile SCA corresponds to those made with machine learning. For this type of attack, a device with encryption algorithms is required, which allows for characterizing the vulnerabilities presented by it and thus obtaining a model.

### 2.2. Non-Profiled SCAs

Non-profiled SCAs require only the encrypted message. There are two major categories for this type of attack: Active Attacks and Passive Attacks.

#### 2.2.1. Active Attacks

When an adversary performs an attack, it somehow exerts an influence over the behavior of the cryptographic device. Then, the attacker can notice a difference in the signals measured on the cryptographic device. From these waves, an attacker can extract secret information.

However, these attacks are considered invasive attacks. In an invasive attack, the attacker physically modifies the cryptographic device. For example, the attacker can induce a signal in the cryptographic device. This signal gives a change in the memory, inside the computation of the data, or the cryptographic algorithm. Fault Attacks (FA) are examples of these kinds of attacks

Due to the nature of these attacks, cryptographic devices are usually irreversibly damaged.

#### 2.2.2. Passive Attacks

On the other hand, passive attacks are not invasive. These kinds of attacks do not interfere with the operation of the device. Hence, the device is not damaged.

Power Analysis Attacks (PAA) [8], Timing Analysis Attacks (TAA) [7], and Electromagnetic Analysis Attacks (EAA) [10] are examples of Passive Attacks.

In Power Analysis Attacks (PAAs), the attacker takes advantage of the fact that the power consumed is correlated to the algorithm and the data processed by the crypto device.

Besides, the PAA can be classified into two categories: Simple Power Analysis (SPA) and Differential Power Analysis (DPA) [8].

It is possible to get the secret key of a cryptosystem using SPA by making one or a few measurements of the power traces of the power consumption when the device is performing its cryptographic operations.

On the other hand, DPA requires many measurements of the power traces of the power consumption. To get these measures, an attacker or researcher uses different input messages on the cryptographic device. A statistical analysis of the measured power traces is carried out. It should be mentioned that DPA is more powerful and more difficult to prevent than SPA.

### 2.3. Countermeasures to Passive Attacks

Many cryptographic algorithms, such as RSA [13], use modular exponentiation as the core. Modular exponentiation uses the binary representation of the exponent. SCAs use the measured traces to find the binary form of the exponent.

Before Kocher published his work, the *Square-and-multiply Algorithm* was the most used for modular exponentiation. This approach has two forms *Left-to-right* (SaML2R) and *Right-to-left* (SaMR2L). Algorithm 1 shows the Square-and-Multiply, Left to Right algorithm.
**Algorithm 1** Square-and-Multiply, Left to Right**Require:** 
*m*, $d={\left(d_{n-1},\cdots ,d_{0}\right)}_{2}$, modulus *N***Ensure:** 
$S=m^{d}\;\mathrm{mod}\;N$
 1:S←1 2:**for***i* from (n−1) down to 0 **do** 3:   $S\leftarrow S^{2}\;\mathrm{mod}\;N$ 4:   **if** ($d_{i}=1$) **then** 5:     S←S·m 6:   **end if** 7:**end for** 8:**return***S*


However, this technique is vulnerable to SPA and Timing Analysis Attacks. The SaM algorithm calculates modular multiplication and square if the exponent bit is 1, as shown in steps 4 and 5 in Algorithm 1; or it processes just one square when the exponent bit is 0.

Coron supplied the first technique to defeat the SPA and Timing Analysis Attack [14]. This regular technique is called *Square-and-multiply Always* (SaMA). Specifically, Algorithm 2 shows the Left to Right version of the SaMA algorithm. This algorithm, in contrast to the *Square and Multiply Algorithm*, calculates a square followed by a multiplication, independently of the value of the corresponding bit of the exponent. Namely, this algorithm calculates a square followed by a multiplication when the bit of the exponent is 0 or 1.
**Algorithm 2** Square-and-Multiply Always, Left to Right**Require:** 
*m*, $d={\left(d_{n-1},\cdots ,d_{0}\right)}_{2}$, modulus *N*
**Ensure:** 
$S=m^{d}\;\mathrm{mod}\;N$
 1:S←1 2:**for** *i* from (n−1) down to 0 **do** 3:   $R\left[0\right]\leftarrow S^{2}\;\mathrm{mod}\;N$ 4:   R[1]←S[0]·mmodN 5:   $S\leftarrow S[d_{i}]\;\mathrm{mod}\;N$ 6:**end for** 7:**return** 
*S*


Table 1 shows a brief example of the SaMAL2R algorithm execution. In this example, the parameters are $d=97_{10}=1100001_{2}$ as the exponent and *m* as the input message.

As shown in the example, in all the steps the same operations are always carried out. However, this algorithm uses dummy operations during its execution. These dummy operations stand for vulnerability because they can be attacked through DPA.

On the other side, Mayima et al. proposed an algorithm to defeat DPA [15], called Binary Expansion with the Random Initial Point (BRIP) algorithm. Algorithm 3. This algorithm protects the input message with a *random value*, and it does not use *dummy operations* during cryptographic operations.
**Algorithm 3** BRIP**Require:** 
*m*, $d={\left(d_{n-1},\cdots ,d_{0}\right)}_{2}$, modulus *N*
**Ensure:** 
$S=m^{d}\;\mathrm{mod}\;N$
 1:S[0]←r 2:$S\left[1\right]\leftarrow r^{-1}$ 3:$S\left[2\right]\leftarrow m\cdot r^{-1}$ 4:**for** *i* from (n−1) down to 0 **do** 5:   $S\left[0\right]\leftarrow S{\left[0\right]}^{2}\;\mathrm{mod}\;N$ 6:   **if** $d_{i}=0$ **then** 7:     S[0]←S[0]·S[1]modN 8:   **else** 9:     S[0]←S[0]·S[2]modN 10:   **end if** 11:**end for** 12:S←S[0]·S[1]modN 13:**return** 
*S*


Table 2 shows a brief example of the execution of the BRIP algorithm. In this example, the parameters are $d=97_{10}=1100001_{2}$ as the exponent and *m* as the input message.

Nevertheless, this algorithm requires the calculation of random numbers for its execution, which stands for excessive consumption during implementation in programmable devices such as FPGA.

The previously mentioned algorithms were attacked by a special type of SCA.

## 3. N−1Attack

There are many kinds of passive attacks. An engrossing attack within passive attacks is the *Chosen-message Attack*. The *Chosen-message Attack* uses a specific input message. This input message produces a particular behavior during the performance of an algorithm. One of these kinds of attacks was developed by Yet et al. [16] and implemented by Miyamoto et al. [17]. This attack is called N−1Attack. This technique is used against the modular exponentiation of the SaMA and BRIP algorithms.

This N−1Attacks works, considering the two following observations in the RSA context with modulus *N*:The first observation is that ${\left(N-1\right)}^{2}\equiv 1\;\mathrm{mod}\;N$, this is extended to the fact that ${\left(N-1\right)}^{j}\equiv 1\;\mathrm{mod}\;N$ for any even integer *j*;The second observation is similar to the earlier one when ${\left(N-1\right)}^{k}\equiv \left(N-1\right)\;\mathrm{mod}\;N$ for any odd integer *k*.

These two observations in modular exponentiation will give only two values in the intermediate operations: a 1 value when the bit processed of the exponent is 0 and N−1 when the bit is 1. Hence, an attacker will see only two patterns in the power traces of the measurements. Thus, the secret key, being the exponent, can be obtained.

An attacker or researcher may use the N−1Attack against the BRIP algorithm. In general, with this technique, anyone can attack algorithms that use a random value to protect the input message.

### Countermeasures to N−1Attack

In the literature, there are proposals to defeat N−1Attacks. Some of these proposals are mentioned below.

The first proposal is to blind the message. The BRIP algorithm shows this countermeasure. The input message, *m*, is blind using a random value, *r*. A second proposal is to block the special input message. In this countermeasure, the input message is compared to the value N−1 and is closed up when both are equal. However, before the proposals are vulnerable to the Chosen-message Attack, this technique follows the same idea as the N−1Attack.

Finally, D. Tinoco [18] made two proposals to countermeasure the N−1Attack. These countermeasures do not require blinding the message, and they do not have to block the input message when it is N−1.

To calculate modular exponentiation, Tinoco follows the property shown in (Equation 1).
(1)$$m^{a^{\prime }}=m^{2a+b}={\left(m^{a}\right)}^{2}\cdot m^{b}$$

Tinoco used (Equation 1) to apply it in modular exponentiation algorithms. Finally, he presented Equation (Equation 2). This equation is the representation that works with intermediate-even exponents.
(2)$$m^{d}={\left(\cdots {\left({\left(m^{2d_{n-1}}\right)}^{2}\cdot m^{2d_{n-2}}\right)}^{2}\cdots m^{2d_{2}}\right)}^{2}\cdot m^{2d_{1}}\cdot m^{d_{0}}$$

The proposals by D. Tinoco follow the idea of SaMAL2R and BRIP algorithms, and his Equation (Equation 2). Algorithm 4 shows the *Modified square-and-multiply always, left to right*, while Algorithm 5 shows the *Modified BRIP Algorithm*.
**Algorithm 4** Modified Square-and-Multiply Always, Left to Right**Require:** 
*m*, $d={\left(d_{n-1},\cdots ,d_{0}\right)}_{2}$, modulus *N*
**Ensure:** 
$S=m^{d}\;\mathrm{mod}\;N$
 1:S←1 2:M←m·mmodN 3:**for** *i* from (n−1) down to 1 **do** 4:   $R\left[0\right]\leftarrow S^{2}\;\mathrm{mod}\;N$ 5:   R[1]←R[0]·MmodN 6:   $S\leftarrow R[d_{i}]\;\mathrm{mod}\;N$ 7:**end for** 8:**if** 
$d_{0}=0$
 **then**
 9:**    return** 
*S* 10:**else** 11:**    return** 
S·mmodN 12:**end if**


**Algorithm 5** Modified BRIP**Require:** 
*m*, $d={\left(d_{n-1},\cdots ,d_{0}\right)}_{2}$, modulus *N*
**Ensure:** 
$S=m^{d}\;\mathrm{mod}\;N$
 1:S[0]←r 2:$S\left[1\right]\leftarrow r^{-1}$ 3:$S\left[2\right]\leftarrow m\cdot m\cdot r^{-1}\;\mathrm{mod}\;N$ 4:**for** *i* from (n−1) down to 1 **do** 5:   $S\left[0\right]\leftarrow S{\left[0\right]}^{2}\;\mathrm{mod}\;N$ 6:   **if** $d_{i}=0$ **then** 7:     S[0]←S[0]·S[1]modN 8:   **else** 9:     S[0]←S[0]·S[2]modN 10:   **end if** 11:**end for** 12:**if** 
$d_{0}=1$
 **then** 13:   S[0]←S[0]·S[1]·mmodN 14:**else** 15:   S[0]←S[0]·S[1]modN 16:**end if** 17:**return** 
S[0]


Since the algorithms run from n−1 to 1, the last square and the last multiplication by $m^{2d_{0}}$ are eliminated. To compute the correct value, an *if statement* replaces the multiplication by $m^{d_{0}}$.

When the exponentiation is executed by MSaMAL2R or BRIP algorithms, the two patterns are not shown. Conversely, only one regular and masked shape in the power traces is observable.

Although these algorithms are implemented in software by the author, they are not implemented in programmable devices such as FPGAs.

## 4. The Proposed Algorithm

Even when it is possible to implement Algorithm 5 in programmable devices such as FPGAs, it is necessary to generate a *random value*, *r*, and calculate its inverse, $r^{-1}$. At this point, it is possible to create random values in an FPGA. However, a reduction of the resources used by the FPGA is necessary. Moreover, the algorithm requires three registers to store intermediate values used in the loop.

On the other hand, Algorithm 4 does not require generating one random value. This advantage helps saving resources compared with the MBRIP algorithm. As stated in the MBRIP algorithm, three registers are needed to store intermediate values in the loop, going against the compromise to reduce the device’s resources.

It should be remembered that the main operation of modular exponentiation algorithms is *modular multiplication*. Modular multiplication was developed and synthesized in the FPGA.

### 4.1. Montgomery Multiplication

There are several proposals reported in the literature on modular multiplications. P. Montgomery developed an interesting modular multiplication [19]. This proposal, known as Montgomery multiplication, has been implemented in FPGAs [20,21,22]. Besides, there is a technique to improve Montgomery multiplication. This technique avoids the final subtraction of the original algorithm. Algorithm 6 shows the description steps suggested in this work for Montgomery multiplication.
**Algorithm 6** Montgomery Multiplication**Require:**$N={\left(N_{w-1}\cdots N_{1}N_{2}\right)}_{b}$, $x={\left(x_{w-1}\cdots x_{1}x_{0}\right)}_{b}$, $y={\left(y_{w-1}\cdots y_{1}y_{0}\right)}_{b}$ with x,y<2N, $2N<R=b^{w}$ with gcd(N,b)=1, and $N^{\prime }=-N^{-1}\;\mathrm{mod}\;b$
**Ensure:** 
$A=xyR^{-1}\;\mathrm{mod}\;N$
 1:A←0 { with $A={\left(A_{w-1}\cdots A_{1}A_{0}\right)}_{b}\}$ 2:**for** *i* from 0 to (w−1) **do** 3:   $u_{i}\leftarrow \left(A_{0}+x_{i}\cdot y_{0}\right)\cdot N^{\prime }\;\mathrm{mod}\;b$ 4:   $A\leftarrow (A+x_{i}\cdot y+u_{i}\cdot N)/b$ 5:**end for** 6:**return** 
*A*


An engrossing proposal for implementing Montgomery multiplication is called *Systolic Architecture* [9,23]. Systolic Architecture uses regular blocks called *Processing Elements* (PEs), which is an advantage for FPGA implementation.

In this approach, the operands are split into words with size *radix b* because base *b* is in the power of 2 with *k* bits.

Taking this into account, Algorithm 7 shows the version of the Montgomery multiplication for implementation in FPGA.
**Algorithm 7** Montgomery multiplication for FPGA**Require:** 
*N*, *x* and *y* with a length of $b^{w}$ bits, and $N^{\prime }$ of *k* bits
**Ensure:** 
$xyR^{-1}\;\mathrm{mod}\;N$
 1:A←0 2:**for** *i* from 0 to (n−1) **do** 3:   Take the lowest *k* bits of $u_{i}\leftarrow \left(A_{0}+x_{i}y_{0}\right)N^{\prime }$ 4:   **for** *j* from 0 to (w−1) **do** 5:     $\left(c_{j},A_{j}\right)=\left(A_{j}+x_{i}y+u_{i}N+c_{j}\right)\gg k$ 6:   **end for** 7:**end for** 8:**return** 
*A*


The operation ≫ in step 5 of Algorithm 7 means the division by *b* in step 6 of Algorithm 6.

However, Montgomery multiplication has a disadvantage. To get the correct result, the *x* and *y* operands must be translated into the Montgomery domain. The operands translation is executed using the same technique as Montgomery. Then, *x* and $R^{2}\;\mathrm{mod}\;N$ are used for $\tilde{x}$ input and, *y* and $R^{2}\;\mathrm{mod}\;N$ are used for $\tilde{y}$ input. Then, the original values are $\tilde{x}$ and $\tilde{y}$.

An added Montgomery multiplication must be done to retrieve the result from the Montgomery domain. It uses the *A* value and one value of 1 as the inputs. However, it is inexpensive for modular exponentiation. It is done only once during the exponentiation execution.

### 4.2. Proposed Modular Exponentiation

As previously mentioned, Algorithm 4 has some advantages.

First, we consider the requirements for the Montgomery multiplication. Using this way, we propose to change Algorithm 4. That proposal is suitable for implementation on FPGAs. Algorithm 8 shows this new proposal.
**Algorithm 8** Modified Square-and-Multiply Always, Left to Right for FPGA**Require:** 
$N={\left(N_{w-1}\cdots N_{1}N_{0}\right)}_{b}$, $m={\left(m_{w-1}\cdots m_{1}m_{0}\right)}_{b}$, $d={\left(d_{n-1\cdots d_{0}}\right)}_{2}$ with m<2N, $2N<R=b^{w}$ with gcd(N,b)=1, and $N^{\prime }=-N^{-1}\;\mathrm{mod}\;b$
**Ensure:** 
$S=m^{d}\;\mathrm{mod}\;N$
 1:S←RmodN 2:$M\left[0\right]\leftarrow m\cdot R^{2}\;\mathrm{mod}\;N$ 3:M[1]←M[0]·M[0]modN 4:**for** *i* from (n−1) downt to 1 **do** 5:   $R\left[0\right]\leftarrow S^{2}\;\mathrm{mod}\;N$ 6:   R[1]←R[0]·M[1]modN 7:   $S\leftarrow R[d_{i}]\;\mathrm{mod}\;N$ 8:**end for** 9:S←S·M[0]modN 10:S←S·1modN**return** 
*S*


The values RmodN and $R^{2}\;\mathrm{mod}\;N$ are inputs. As it is mandatory in the Montgomery multiplication algorithm, the input values are settled in the Montgomery domain; step 2 sets the message input in the Montgomery domain. The result is retrieved in step 10.

In the context of RSA, the exponent *d* is always an odd number. That is, $d_{0}$ will always be 1; considering this, step 9 in Algorithm 8 has replaced the last condition in Algorithm 4.

Table 3 shows a brief example of the execution of the proposed algorithm. In this example, the parameters are $d=97_{10}=1100001_{2}$ as the exponent and (N−1) as the input message.

Notice that the intermediate values are always the same. The above example proves the correct operation of the proposed technique.

The proposed algorithm requires one extra register compared with the MSaMAL2R algorithm; this is a disadvantage for programmable devices with reduced resources. However, the extra register is not a drawback, due to the number of resources in the FPGA.

## 5. Implementation of the Proposed Algorithm in FPGA

The implementation of modular multiplication is essential because, as previously mentioned, it is the core of modular exponentiation. Thus, the implementation of modular multiplication defines the number of resources used by the FPGA. Thus, an optimal design of the implementation can be achieved.

### 5.1. Implementation of Proposed Modular Multiplication for FPGA

First, we implement the Montgomery multiplication into FPGA. We used the Systolic Architecture with a radix of 16 bits; the operands were chosen accordingly to avoid the final subtraction, as previously mentioned.

With the radix of 16-bits and 1024-bit RSA, we split the operands into 64 words; thus, there are 64 PEs. This operation takes only one clock cycle.

All these PEs are regular, which means that only one PE was developed, and this PE was replicated 64 times.

Figure 1 shows the block diagram of the Montgomery multiplication architecture implemented in FPGA.

The value *x* is *left-shifted* 16 bits, and this value is sent to all PEs and the block called **$u_{i}$**. The task of this last block is to calculate the value of step 3 in Algorithm 7. As previously mentioned, the least significant bits are the only ones considered.

Figure 2 shows the block diagram of the individual PE implemented. In the first PE, the *input-carry value* was set to zero. In the last PE, the *output carry value* was used as the final $A_{w-1}$.

We use Digital Signal Processors (DSPs) embedded in the FPGA to reduce resources for the implementation of modular multiplication. These DSPs have internally one pre-adder, one 18×25 bits multiplier, and an accumulator. All PEs were set in a pipeline fashion. The implementation of the DSPs, which we use for the multiplier and the pre-adder, is carried out as recommended by the manufacturer Xilinx in its documentation.

The pre-adder receives operands from the multiplexers 2 and 3, as shown in Figure 2. The inputs of multiplexer 2 are the values $y_{j}$ and $u_{i}$. The inputs of the multiplexer 3 are the values $A_{j}$ and the carry of the earlier PE, $carry_{j-1}$. The inputs of the multiplier are the values from the multiplexer 1 and the pre-adder. The accumulator of the DPS adds the earlier result with the value given for the multiplier. The purpose of this accumulator is to perform the additional operation given in step 5 of Algorithm 7.

A Finite State Machine (FSM) controls the operations of the Systolic Architecture. This FSM has five states to manage all PEs and to set the corresponding values into each PE.

Although five states are used and *one state consumes one clock cycle*, the first one is for idling, and the last state is for resetting. Thus, *three clock cycles are used by the algorithm for performing all the operations required to perform the modular multiplication*. So, considering that all the 64 PEs and each PE used three clock cycles for processing the data, plus one clock cycle to set up the beginning of the process and one to return the final result, the total clock cycles running for the modular multiplication is 194.

In the present work, we used a clock of 100 MHz for different platforms. Each PE uses three clock cycles, so the latency of each PE is 30 ns. Considering that we have 64 PEs, we have 1920 ns to operate. However, there are two extra clock cycles, one for idling and one for resetting, then the time for executing one modular multiplication is 1940 ns.

### 5.2. Implementation in FPGA of the Proposed Modular Exponentiation

For the implementation of the modular exponentiation proposed, an FSM, which follows Algorithm 8, was used. Figure 3 shows the block diagram of the proposed method.

All registers were set according to 1024-bit RSA and eight registers of 1024-bits were used for this implementation distributed as follows: one register to store the input message *m*; one register to store the value RmodN; one register to store $R^{2}\;\mathrm{mod}\;N$; two more registers to store the values of ***M***; two registers to store the intermediate values ***R***; and one register to store the output *S*.

External RAM blocks were not used to implement the proposed modular exponentiation. Instead of using an external block of RAM, registers were used to store the values.

The FSM controls the steps within the algorithm. This FSM passes the data into all registers of the block Montgomery multiplication. Passing the data into all registers is done considering the *d value* exponent, as was shown in Algorithm 8.

Figure 3 shows the block Montgomery multiplication.

Then, the output is stored in the register labeled *S* in the block diagram. In this proposal, the FSM saves the output value in a register instead of in an external RAM block.

## 6. Results

We used two different families of FPGAs. For the first one, we use a development board Nexys 4 DDR of Digilent^®^. The FPGA used is the Artix XC7A100T-CSG324 of Xilinx^®^, and it was implemented and synthesized using Vivado 2022.1. For the last one, we used the FPGA Virtex-6 XC6VLX75T-2FF484, which was only simulated using the post place-and-route results. For the synthesized one, we used ISE 14.7 Web Pack Edition. In both cases, we used a clock of 100 MHz.

One of the purposes of carrying out the implementation of our proposal using two different families of FPGAs is not only to compare it with other architectures, but also to show that the results obtained are similar in consumption and speed, as well.

Table 4 shows all the resources used for the implementation of the modular exponentiation, with N=1024 bits, proposed in this work using the two families of FPGAs mentioned previously.

In the case of Artix-7, only 9803 Look-Up Tables (LUT) and 10,568 Flip-Flops (FF) were used, which is only 15.46% and 8.33% with respect to the total resources available. However, the most important parameter is the number of DSPs in both platforms. Both implementations used 66 DSPs.

These resources also include those used by the implementation of the Montgomery multiplication. These results show that only part of the FPGA resources is used for modular exponentiation.

Table 5 shows the resources used by modular exponentiation and Montgomery multiplication in the case of Artix-7.

Table 5 shows that the implementation of the modular multiplication used 9714 LUT, hence, only 89 extra LUT were needed for the implementation of the modular exponentiation. Similarly, the modular multiplication used 5338 FF, so, 5231 extra FF were needed for the implementation of the modular exponentiation.

As was previously mentioned, the Montgomery multiplication takes 194-clock cycles. Modular multiplication is performed by the proposed modular exponentiation. Algorithm 8 takes 2052 modular multiplications. The total multiplication takes 398,088 clock cycles. However, step 7 of Algorithm 8 requires 1 clock cycle to store the data into *S*. Besides, the entire loop of encryption or decryption of data takes a total of 399,113 clock cycles.

The value of 399,113 clock cycles is for the modular exponentiation. Nevertheless, the number of clock cycles needed to store and read the data from external communication was not considered.

Table 6 shows a comparison of the State-of-the-Art Modular Exponentiation Architectures. The exponent has a length of 1024 bits in all implementations.

Table 6 compares the different architectures used to perform modular exponentiation, which includes different platforms. All of them have their advantages and disadvantages. The results show that our work has a reduction in the number of LUT compared with the rest of the works.

Our architecture, using Virtex-6, has a reduction of 3550 LUTs compared to Ref. [24] and 115,528 compared to Ref. [21]. In the case of Artix-7, there is a reduction of 4.31% of LUTs compared to Ref. [9].

The present work requires a smaller number of FFs. In the case of Virtex-6, there is a reduction of 62,583 of FFs compared to Ref. [21]. In the case of Artix-7, it requires a small increase of FFs, only of 1.38% compared to Ref. [9].

An advantage of the present work is that it does not require the use of memory blocks (BRAM), in contrast with the rest of the works, since they require a large number of memory blocks. For Virtex-6, the work presented in Ref. [24] requires 32 blocks of 32 kb and 2 blocks of 18 kb; for the case of the work presented in Ref. [21] it requires 132 BRAMs from the total that the FPGA has. For Artix-7, our architecture does not require the use of BRAM, while the work presented by Somnath requires 7.81% of BRAM.

Our proposal, in the case of the Virtex-6 platform, requires a slightly larger number of slices, using only 779, compared with those presented in Ref. [24], however, it has a reduction of 32,885 slices compared with the architectures presented in Ref. [21].

As previously mentioned, the number of DSPs is an important parameter to take into account. Our proposed implementation used only 66 DSP.

The latency or the number of clock cycles that our architecture takes to perform modular exponentiation is greater than those shown in other works [21,24]. However, our proposal shows a reduction in the number of resources.

Finally, earlier results show a great reduction in resources used for both families of FPGA. The foregoing, together with the proposal made in Algorithm 8, produces an increase in security.

## 7. Discussion

In the present work, we developed an implementation of the MSaMAL2R for FPGA, which is effective against non-profiled side channel attacks.

This implementation requires less consumption of resources used by the FPGA for its operation, as mentioned in the previous section. However, it requires more clock cycles to perform the modular exponentiation. To speed-up the performance, it is necessary to improve the architecture of the Montgomery modular multiplication presented in this work.

Our proposal not only requires a smaller number of DSPs but also a smaller number of slices. This reduction in resources is helpful due to the lower energy consumption during its operation, which makes it less susceptible to non-profiled SCA.

Although it has shown a reduction in resources for the encryption or decryption of information, in addition to being less vulnerable to side-channel attacks, it is necessary to study other types of attacks on this implementation. Some of these attacks may be profiled SCA.

Because we use VHDL for the hardware description, it is possible to scale or use the same code in other types of FPGAs from the same manufacturer, however, it can also be used in FPGAs from other manufacturers with the same characteristics as the FPGA used in the present work. To support the above statement, we implemented the same proposal in two different FPGAs families using the same VHDL code. The first one used a high-cost FPGA such as the Virtex and the second one implemented a low-cost FPGA such as the Artix.

## 8. Conclusions

This work presents a proposal to defeat the N−1Attack without cost penalties. One extra register is required, and no generations of *random values* are needed. Thus, it allows reducing the complexity of the application and reduces the resources in the implementation.

The resources shown in the results section demonstrated that the proposed implementation is a worthy possibility to be implemented in hardware using programmable devices such as FPGA. The proposal made in this paper shows a reduction in resource consumption; specifically, a consumption of LUTs by 3550 LUTs and 62,583 of FFs.

As was mentioned previously, we implemented our proposal using two different families of FPGAs. These two families are Virtex-6 and Artix-7. The former is one of the costliest in price and the last is one of the cheaper options. Both are from the Xilinx family. The implementation of our proposal in both types of FPGA presents similar results in terms of the number of resources used. This shows that the proposal can be implemented in low-cost and low-resource devices, as was the case with Artix-7. This allows it to be used in portable devices such as those used in IoT. It even allows other approaches such as Neural Networks to be implemented in the case of using high-resource FPGAs such as Virtex.

The running time of the modular exponentiation mostly depends on the modular multiplication. Even though there are implementations of modular multiplication with a faster running time than this proposal, nevertheless, the resources needed are higher than in our work.

Table 6 shows different techniques to implement modular exponentiation. All of them are designed to defeat SCAs. However, our proposal has the advantage that fewer resources are needed. This is an advantage over the rest, thus increasing the security due to requiring less power consumption. 

## Figures and Tables

**Figure 1 micromachines-13-01872-f001:**
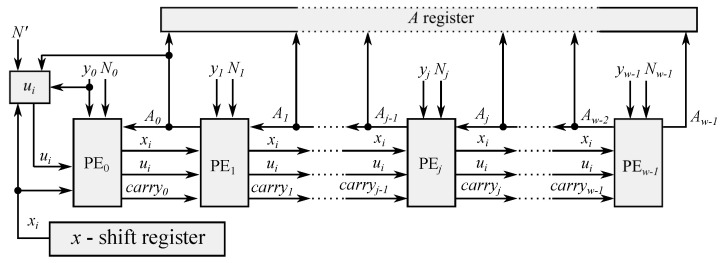
Block diagram of the systolic architecture for Montgomery multiplication in FPGA.

**Figure 2 micromachines-13-01872-f002:**
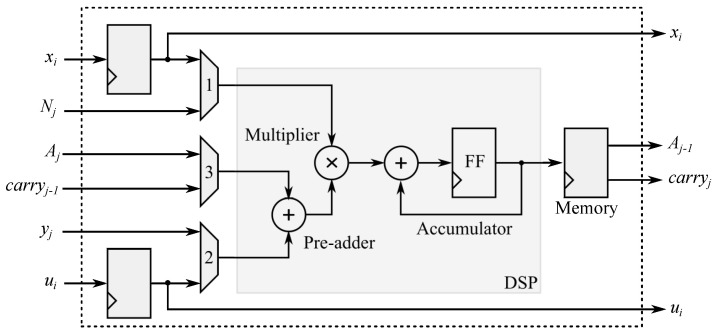
Block diagram of a processing element in FPGA.

**Figure 3 micromachines-13-01872-f003:**
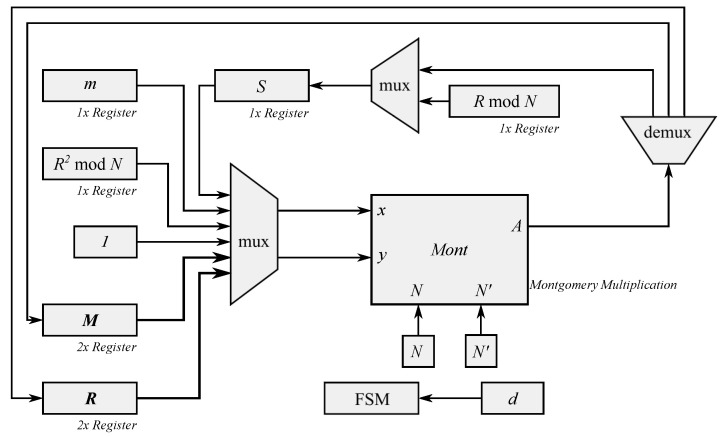
Block diagram of the proposed Modular Exponentiation.

**Table 1 micromachines-13-01872-t001:** Example of the execution of the Algorithm SaMAL2R.

*i*	$d_{i}$	*R Values*	*Result*
6	1	$R\left[0\right]=S^{2}\;\mathrm{mod}\;N=1\;\mathrm{mod}\;N$ R[1]=S[0]·mmodN=mmodN	S=mmodN
5	1	$R\left[0\right]=m^{2}\;\mathrm{mod}\;N=m^{2}\;\mathrm{mod}\;N$ $R\left[1\right]=m^{2}\cdot m\;\mathrm{mod}\;N=m^{3}\;\mathrm{mod}\;N$	$S=m^{3}\;\mathrm{mod}\;N$
4	0	$R\left[0\right]={\left(m^{3}\right)}^{2}\;\mathrm{mod}\;N=m^{6}\;\mathrm{mod}\;N$ $R\left[1\right]=m^{6}\cdot m\;\mathrm{mod}\;N=m^{7}\;\mathrm{mod}\;N$	$S=m^{6}\;\mathrm{mod}\;N$
3	0	$R\left[0\right]={\left(m^{6}\right)}^{2}\;\mathrm{mod}\;N=m^{12}\;\mathrm{mod}\;N$ $R\left[1\right]=m^{12}\cdot m\;\mathrm{mod}\;N=m^{13}\;\mathrm{mod}\;N$	$S=m^{12}\;\mathrm{mod}\;N$
2	0	$R\left[0\right]={\left(m^{12}\right)}^{2}\;\mathrm{mod}\;N=m^{24}\;\mathrm{mod}\;N$ $R\left[1\right]=m^{24}\cdot m\;\mathrm{mod}\;N=m^{25}\;\mathrm{mod}\;N$	$S=m^{24}\;\mathrm{mod}\;N$
1	0	$R\left[0\right]={\left(m^{24}\right)}^{2}\;\mathrm{mod}\;N=m^{48}\;\mathrm{mod}\;N$ $R\left[1\right]=m^{48}\cdot m\;\mathrm{mod}\;N=m^{49}\;\mathrm{mod}\;N$	$S=m^{48}\;\mathrm{mod}\;N$
0	1	$R\left[0\right]={\left(m^{48}\right)}^{2}\;\mathrm{mod}\;N=m^{96}\;\mathrm{mod}\;N$ $R\left[1\right]=m^{96}\cdot m\;\mathrm{mod}\;N=m^{97}\;\mathrm{mod}\;N$	$S=m^{97}\;\mathrm{mod}\;N$

**Table 2 micromachines-13-01872-t002:** Example of the execution of the Algorithm BRIP.

*i*	$d_{i}$	*R Values*	*Result*
6	1	$S\left[0\right]=r^{2}\;\mathrm{mod}\;N$	S[0]=r·mmodN
5	1	$S\left[0\right]=r^{2}\cdot m^{2}\;\mathrm{mod}\;N$	$S\left[0\right]=r\cdot m^{3}\;\mathrm{mod}\;N$
4	0	$S\left[0\right]=r^{2}\cdot m^{6}\;\mathrm{mod}\;N$	$S\left[0\right]=r\cdot m^{6}\;\mathrm{mod}\;N$
3	0	$S\left[0\right]=r^{2}\cdot m^{12}\;\mathrm{mod}\;N$	$S\left[0\right]=r\cdot m^{12}\;\mathrm{mod}\;N$
2	0	$S\left[0\right]=r^{2}\cdot m^{24}\;\mathrm{mod}\;N$	$S\left[0\right]=r\cdot m^{24}\;\mathrm{mod}\;N$
1	0	$S\left[0\right]=r^{2}\cdot m^{48}\;\mathrm{mod}\;N$	$S\left[0\right]=r\cdot m^{48}\;\mathrm{mod}\;N$
0	1	$S\left[0\right]=r^{2}\cdot m^{96}\;\mathrm{mod}\;N$	$S\left[0\right]=r\cdot m^{97}\;\mathrm{mod}\;N$
* **Retrieve result** *			
		$S\left[0\right]=r\cdot m^{97}\cdot r^{-1}\;\mathrm{mod}\;N$	$S\left[0\right]=m^{97}\;\mathrm{mod}\;N$

**Table 3 micromachines-13-01872-t003:** Example of the execution of the proposed algorithm.

*i*	$d_{i}$	*R Values*	*Result*
6	1	$R\left[0\right]=R^{2}\;\mathrm{mod}\;N=R\;\mathrm{mod}\;N$ R[1]=R·RmodN=RmodN	S=RmodN
5	1	$R\left[0\right]=R^{2}\;\mathrm{mod}\;N=R\;\mathrm{mod}\;N$ R[1]=R·RmodN=RmodN	S=RmodN
4	0	$R\left[0\right]=R^{2}\;\mathrm{mod}\;N=R\;\mathrm{mod}\;N$ R[1]=R·RmodN=RmodN	S=RmodN
3	0	$R\left[0\right]=R^{2}\;\mathrm{mod}\;N=R\;\mathrm{mod}\;N$ R[1]=R·RmodN=RmodN	S=RmodN
2	0	$R\left[0\right]=R^{2}\;\mathrm{mod}\;N=R\;\mathrm{mod}\;N$ R[1]=R·RmodN=RmodN	S=RmodN
1	0	$R\left[0\right]=R^{2}\;\mathrm{mod}\;N=R\;\mathrm{mod}\;N$ R[1]=R·RmodN=RmodN	S=RmodN
* **Last bit** *			
0	1	S=S·M[0]modN	S=M[0]modN
* **Retrieve result** *			
		S=S·1modN	S=(N−1)modN

**Table 4 micromachines-13-01872-t004:** Resources used by the FPGA for the proposed Modular Exponentiation.

Device Utilization				
Platform	LUTs	FFs	Slices	DSPs
Virtex-6	7834/46,560	13,286/93,120	4249/11,640	66/288
Artix-7	9803/63,400	10,568/126,800	4485/15,850	66/240

**Table 5 micromachines-13-01872-t005:** Resources used by the FPGA for the proposed Modular Multiplication.

Device Utilization				
Operation	LUT (63,400)	FF (126,800)	Slices (15,850)	DSP (240)
Exponentiation	9803	10,568	4485	66
Multiplication	9714	5337	3970	66

**Table 6 micromachines-13-01872-t006:** Performance comparison of the State-of-the-Art Modular Exponentiation with N=1024.

Work	Platform	Architecture	LUT	FF	BRAM	Slices	DSPs	Freq (MHz)	Latency (Cycles)
Wangchen [24]	Virtex-6	FMLE	11,834	-	32/2 ^1^	3470	18	-	822 per FMLM
Wangchen [24]	Virtex-6	FMLE protected	11,884	-	32/2 ^1^	3433	18	-	822 per FMLM
Vankatesh [21]	Virtex-7	Pre-computation (Montgomery)	128,814 (42.43%)	75,839 (12.49%)	132 ^2^ (12.82%)	37,134 (48.92%)	-	200	14,813
Vankatesh [21]	Virtex-7	Pre-computation (Montgomery)	128,814 (42.43%)	75,839 (12.49%)	132 ^2^ (12.82%)	37,134 (48.92%)	-	200	14,813
Proposed architecture	Virtex-6	Systolic	7834 (16%)	13,286 (14%)	0%	4249 (36%)	66	100	399,113
Somnath [9]	Artix-7	Systolic	19.77%	6.95%	7.81%	-	-	90.9	-
Proposed architecture	Artix-7	Systolic	9803 (15.46%)	10,568 (8.33%)	0%	4485 (28.3%)	66	100	399,113

^1^ 32 blocks of 32 kb and 2 blocks of 18 kb. ^2^ 132 blocks of the total. - Not mentioned.

## Data Availability

Not applicable.

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
