# Peer review of "SCA-Safe Implementation of Modified SaMAL2R Algorithm in FPGA"

_micromachines, 2022, doi:10.3390/mi13111872_

Round 1

Reviewer 1 Report

Authors have presented a modular exponentiation algorithm implemented in the FPGA to improve the security of operation due to an attack. But the following key issues are to be addressed in order to improve the strand of the presented work.

The authors need to clarify how 194 clock cycles are required for processing 64 PEs provided each PE requires 3 clock cycles.

Even though authors claim to use DSP blocks for implementing PE, it is worth how  the multiplier and adders of the PE are implemented in the FPGA?

What is the latency of a PE?

What is the frequency of fetching the input data considering the delay in the multiplier and adder?

What is the purpose of using both the accumulator and the memory in the implementation of PE?

How many clock cycles are needed to finally fetch the data from the memory?

What is the depth of the memory?

How about the scalability of the proposed algorithm on the other types of FPGAs?

The results presented in table 4 does not make sense as the algorithms are implemented in different FPGAs and with a different frequency of operation. So how to compare the latency, the number of slices, DSP blocks of the proposed algorithm with other algorithms? Suggest authors use a uniform FPGA platform for comparison.

The abstract has to clearly present the qualitative results of the presented work. 

Author Response

Dear reviewer:

We want to thank you for the time you have destined to review of our article, and for the comments you have made about it. These comments will give the article a higher quality.

Point 1: “The authors need to clarify how 194 clock cycles are required for processing 64 PEs provided each PE requires 3 clock cycles.

We clarify where the number comes from, this was done in the subsection of Implementations of proposed modular multiplication for FPGA

Point 2: “… it is worth how the multiplier and adders of the PE are implemented in the FPGA?

We added a note on how the multiplier and pre-adder were implemented.

Point 3: “What is the latency of a PE?”

In the work the latency of the PE is set up, however, we expand the explanation.

Point 4: “What is the frequency of fetching the input data considering the delay in the multiplier and adder?”

The delay of the various components is stated in this article; however, we have added a more detailed explanation on this point.

Point 5: “What is the purpose of using both the accumulator and the memory in the implementation of PE?”

We added a brief explanation about how we use the accumulator and the memory.

Point 6: “How many clock cycles are needed to finally fetch the data from the memory?

In the present work, we do not consider the reading and writing of the input and output data. However, we added a note about this point.

Point 7: “What is the depth of the memory?”

As mentioned in the article, we do not use internal memory, we use registers instead. However, we expand the explanation on this point.

Point 8: “How about the scalability of the proposed algorithm on the other types of FPGAs?”

We added a note in the discussion section on this point. Where we clarify the use of other FPGAs.

Point 9: “The results presented in table 4 does not make sense as the algorithms are implemented in different FPGAs and with a different frequency of operation. So how to compare the latency, the number of slices, DSP blocks of the proposed algorithm with other algorithms? Suggest authors use a uniform FPGA platform for comparison.

Thank you for the suggestion. We compare the present work with those previously published, however, we did not find publications that use the same FPGA. But we add several comments in the comparison of the FPGA used in this work with the earlier ones.

Point 10: “The abstract has to clearly present the qualitative results of the presented work.”

We update the abstract.

Reviewer 2 Report

The paper introduced a modified algorithm that deals with modular exponentiation and its implementation to protect information from attacks. However, multiple points were missing, and understanding the proposed approach is challenging; here are the following points to be considered:

  • In the introduction, the authors mentioned: "This paper deals with modular exponentiation and its implementation to protect information from some of the known attacks." which attacks????
  • The introduction section needs more details regarding research problems and the main paper's contribution. 
  • The explanations of algorithms 1 and 2 were missing, making it challenging to follow up and be aware of what algorithms 3 and 4 would like to address. 
  • There is no sufficient explanation for algorithms 3 & 4. 
  • Tables 2 and 3 mainly present the results of the proposed work, but the explanation of both are missing.
  • Similarly, Table 4 compares the proposed approach and the SotA algorithms; however, due to the lack of an explanation, I found it difficult to understand which of these approaches is better than the others and why?? 
  • There is no future vision regarding the paper's work.  

Some other remarks:

  • Too many keywords are used in the paper; please select the most focused keywords due to the paper's objective. 
  • Try to reduce the spaces between the paper paragraphs. 
  • Tables 2 and 3 have the same legends; please update that.

Author Response

Dear reviewer:

We want to thank you for the time you have destined to review of our article, and for the comments you have made about it. These comments will give the article a higher quality.

Point 1: “In the introduction, the authors mentioned: "This paper deals with modular exponentiation and its implementation to protect information from some of the known attacks." which attacks?”

We added some examples in the present work.

Point 2: “The introduction section needs more details regarding research problems and the main paper's contribution”

Thank you for this suggestion. We added more details in the introduction section.

Point 3: “The explanations of algorithms 1 and 2 were missing, making it challenging to follow up and be aware of what algorithms 3 and 4 would like to address”

We added a detail explanation and some examples about algorithms 1 and 2. Moreover, we added an algorithm to clarify the following ones. However, algorithms 3 and 4, works in equivalent manner.

Point 4: “There is no sufficient explanation for algorithms 3 & 4”

We hope that with the added notes about the algorithms mentioned in these points it will be clear how they work.

Point 5: “Tables 2 and 3 mainly present the results of the proposed work, but the explanation of both are missing”

We added a detail explanation about these tables.

Point 6: ”Similarly, Table 4 compares the proposed approach and the SotA algorithms; however, due to the lack of an explanation, I found it difficult to understand which of these approaches is better than the others and why?”

Like the earlier point, we added a detail explanation.

Point 7: “Too many keywords are used in the paper; please select the most focused keywords due to the paper's objective”

We reduce the number of keywords.

Point 8: “Try to reduce the spaces between the paper paragraphs”

Thank you for the suggestion. However, we did not do it. This is difficult due to the template provided by MDPI.

Point 9: “Tables 2 and 3 have the same legends; please update that”

Tables 2 and 3 have different legends, however we understand that it is confusing due to the similarity of the legends. To solve this situation, we emphasize the differences in both legends.

Round 2

Reviewer 1 Report

The rebuttal has to be properly written and presented. For each question at least state the page number and the column number for easy referencing. Some of the important concerns raised are not addressed adequately. For example, consider the following

“The results presented in table 4 does not make sense as the algorithms are implemented in different FPGAs and with a different frequency of operation. So how to compare the latency, the number of slices, DSP blocks of the proposed algorithm with other algorithms? Suggest authors use a uniform FPGA platform for comparison.

It is not clear if the authors understand the question raised. It is not possible to compare the resources of different FPGAs upon implementing the same algorithm as each FPGA family has different resources. 

Another clear example is that the abstract did not state or highlight the quantitative results even though this has been stated in the first review. 

So I suggest the authors to properly address the rebuttal for further considerations. 

Author Response

Thanks for the comments, following are the comments about the corresponding observation.

Up to the authors’ knowledge, after a deep review of the literature related to the implementation of the algorithm in FPGAs, the authors found only one work about the use of similar FPGA platform.

So, we understand the concern of making a comparison of the resources used, although table 4 presents a relationship between the performance and percentage of resources used by the different platforms. The authors think that the proposal here presented gives an approach that contributes to expanding the state of the art in using FPGAs for encryption tasks. It might be interesting to have at hand different options from which people interested in this research area, can choose that which can fit the particular purposes.

Reviewer 2 Report

none

Author Response

Thank you very much for your comments. These have helped to improve the work presented.

A review of the style and language used in this work has been carried out.

Round 3

Reviewer 1 Report

The authors refused to compare their proposed algorithm on different FPGA platforms. Therefore, their following claims stand meaningless. As their claims are important to the presented method,  the scientific merit of this paper is not clearly proven.

Table 6 compares different architectures to perform modular exponentiation, which 309 includes different Platforms and the same Platform (Artix-7). All of them has advantage 310 and disadvantages. 311 The results shown in the Table 6 show that our proposal requires a much smaller 312 number of slices compared with different platforms, it only requires 1,452 slices which is 313 lower compared with the 3,433 slices required for the architecture presented in [23] or the 314 7,012 slices required for the architectures presented in [17].

The presented proposal shows a reduction in the number of LUTs. Although the 316 present work requires a greater number of FF, the BRAM is not used. In addition, our work 317 can operate at a higher frequency. All of the above is compared with the work presented in 318 [24]. 319 As was previously mentioned, the number of DSP is an important parameter. In 320 the implementation, only 66 DSPs are used, compared to previous works, a significant 321 reduction of this amount is shown. 322 According to table 2, the latency or the number of cloc cycles the algorithm takes to 323 perform the modular exponentiation of our proposal is similar to those shown in the works 324 [17] and [23]. However, our proposal shows a reduction of 92,407 clock cycles. This result 325 shows that our proposal is faster in executing the modular exponentiation. 326 Finally, previous results show not only a reduction in resources used but also an 327 increase in processing speed. The foregoing, together with the proposal made in Algorithm 328 8, produce an increase in security.

Author Response

Dear reviewer,

First, thank you very much for your comments. We have taken them into account to improve this work.

We carry out the implementation of our proposal with two families of FPGA. Furthermore, we have compared it with the published works that use the same families of FPGA. This has resulted in an update in Table 6. In the following paragraphs, the explanation of the importance of our work is given, comparing it with said published works.

In addition, Table 4 has been updated, where we show the resources used in the post-synthesis of the designs in the two FPGA families used.

Round 4

Reviewer 1 Report

All the comments raised were addressed. Thank you.